# Effects of Cognitive Sensory Motor Training on Lower Extremity Muscle Strength and Balance in Post Stroke Patients: A Randomized Controlled Study

Kyung-Hun Kim [1] and Sang-Hun Jang [2,*]

1   Department of Physical Therapy, Gimcheon University, 214, Daehak-ro, Gimcheon 39528, Korea; huni040@naver.com
2   Department of Physical Therapy, Korea National University of Transportation, 61, Daehak-ro, Jeungpyeong-gun 27909, Chungcheongbuk-do, Korea
*   Correspondence: upsh22@hanmail.net; Tel.: +82-43-820-5208

**Abstract:** Background: Sensory motor impairment, the most common neuromuscular condition in stroke patients, often contributes to muscle weakness and imbalance. Objective: The purpose of this research was to investigate the effects of cognitive sensory-motor training (CSMT) on the muscle strength and balance ability in post-stroke patients. Methods: Thirty-five participants after stroke were randomly assigned to the CSMT ($n = 17$) or control group ($n = 18$). All participants received 30 min of training each time, five times per week, for six weeks. Lower extremity muscle strength of tibialis anterior (LEMTA) was evaluated using a digital muscular dynamometer. The Medical Research Council (MRC) scale was used to evaluate muscle strengths of the hip joint, knee joint, and ankle joint. For balance ability test, the center of pressure (COP) movement distance and limited of stability (LOS) were measured using BioRescue. Results: LEMTA, MRC scale, balance ability were significantly more improved in the CSMT group than in the control group ($p < 0.05$). Conclusions: Our findings indicate that CSMT is beneficial and effective for improving muscle strength of the lower extremity and balance ability of post-stroke patients.

**Keywords:** cognitive sensory motor; muscle strength; postural balance; stroke





## 1. Introduction

Although functional limitations and disability aspects caused by stroke are different depending on the area and degree of damage, stroke patients generally show problems such as cognitive disorder, motor and sensory impairments and lower extremity dysfunction [1]. As a result, they suffer difficulty in postural control, balance impairments, stabilization of the body against gravity, and disorientation accompanied by abnormal movements due to muscular weakness [2].

CSMT was first proposed by Professor Carlo Perfetti. It is widely known in the rehabilitation program as Perfetti's Method [3,4]. It is a special and comprehensive rehabilitation program that retrains cooperative and systematic guidance and sensory-induced movement control [5]. This method can be used to develop the ability to organize factors for spatial and temporal intensity of the exercise sequence required for interaction between the body and the environment to the maximum. It can be used as an approach to create the preparation stages for the central nervous system to effectively perform various actions or behaviors [6]. Perfetti's CSMT focuses on sensory retraining. It is a method of treatment with particular emphasis on the recognition of a specific position of the joint [5]. It has been reported that training of a multi-component cognitive rehabilitation program for patients with mild cognitive disorder is effective in helping patients perform activities of daily living and improve concentration [7]. Additionally, internal and external sensory input signals have an advantage in motor rehabilitation [8].

In general, methods used in neurorehabilitation are widely used, such as transcranial direct current stimulation [9], functional electrical stimulation [10], treadmill [11], and sensory integration [12] have also conducted. Studies on the effects of sensory-motor training have been reported. Studies that combine cognitive process therapy with various methods such as action observation [13], mirror therapy [14], and motor image [15] have also been actively conducted. Buccino et al. [16] have reported that motor imitation is a cognitive process that includes action observation, motor imagery, motor execution training process. Based on published papers about sensory-motor training, Lynch et al. [17] have trained 21 stroke patients for light touch sensation, discriminative sensation, and proprioception in sitting and standing positions with closed eyes for two weeks and found significant improvement in postural control and balance ability. Geiger et al. [18] have shown improvement in balance ability through a visual feedback exercise program for improving the balance and mobility of stroke patients.

Recent studies on improvement of function have reported that motor function is closely related to cognitive function [19]. The purpose of CSMT based on learning theory is to induce recovery of motor function and enable patients to activate their cognitive process, leading to extensive recovery from damage [3]. Applying only cognitive-motor and cognitive-motor plus other components to the elderly can lead to significant differences in physiological profile assessment, TUG, postural sway, and step reaction time [20]. It has been reported that activation occurs at the primary sensory motor cortex in damaged lesions of the brain after Perfetti's Method training based on FMRI measurements [21]. On the basis of these effects, CSMT has been proven to be an effective method for sensory motor training in many papers published in international journals. Although there are many papers applying CSMT to the upper extremity for cognitive sensory rehabilitation, papers applying it to the lower extremity have not yet been published. Also, the benefits of CSMT improving muscle strength of the lower extremity and balance ability of stroke patients are not so clear. Thus, the purpose of this study was to investigate effects of CSMT on muscle strength of the lower extremity and balance ability of post-stroke patients.

## 2. Materials and Methods

### 2.1. Participants

This study was designed as a single blind randomized study. The subjects of this study included 35 hospitalized patients with stroke who were undergoing physical therapy at K hospital located in Gyeonggi-do. Inclusion criteria were: (1) those who could walk for at least 10 m, (2) those who had a stroke over 6 months ago, (3) those who could communicate, (4) Mini Mental State Examination-Korea (MMSE-K) score was above 24, (5) Brunnstrom recovery stage 2 to 4, (6) those who did not have a problem with walking due to other diseases other than stroke, (7) those with sensory defect in the lower extremity, and (8) those who had voluntarily signed an informed consent form prior to the this study. Exclusion criteria were: (1) vestibular organ or cerebellum-related disease, (2) visual impairment or hearing defect, (3) severe cognitive decline and aphasia, (4) those who had difficulty in conducting the study and those who had hemineglect.

### 2.2. Procedure

This study had an assessor-blinded and randomized controlled trial design. In accordance with the Helsinki Declaration, this study was conducted after obtaining approval from the Institutional Review Board of Sahmyook University. The consent number for bioethics study is 2-1040781-AD-N-01-2016009HR.

Thirty-nine patients were selected based on the inclusion criteria and divided into a CSMT group (*n* = 19) and a control group (*n* = 20) according to the purpose of this study. Groups were selected by randomization using a computer to minimize selection bias (www.randomizer.org, accessed on 14 September 2021). During the study period of 6 weeks, two subjects from the CSMT group and two subjects from the control group

did not receive allocated intervention. Thus, 35 subjects finally participated in this study (Figure 1).

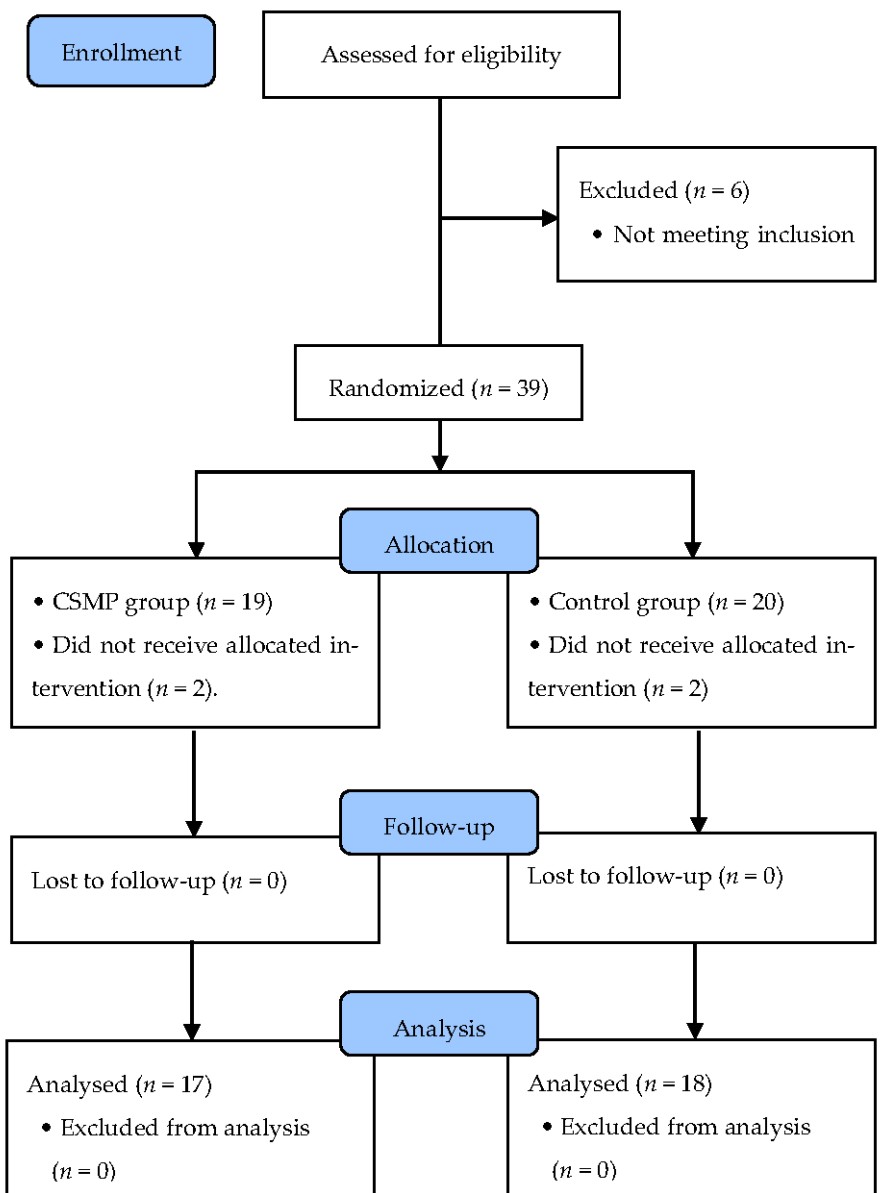

**Figure 1.** Flow diagram of the study.

A pretest was conducted before training and a post-test was carried out after 6 weeks of training. In this study, CSMT and conventional physical therapy were performed in two hospitals: B hospital, 8 subjects in the CSMT group and 10 subjects in the control group; K hospital, 9 subjects in the CSMT group and 8 subjects in the control group. To minimize the effect of CSMT, researchers and therapists conducted experiments after sufficient education and discussion about treatment methods. In the CSMP group, 30 min of CSMT and 30 min conventional physical therapy were carried out 5 times a week. In the control group, conventional therapy was performed for 30 min each time, twice per day. In both hospitals, conventional physical therapy was performed in at the same way. The training was stopped if the training could not be maintained for 30 min depending on the subject's motor ability. Five minutes of rest were allowed if subjects felt fatigue or complained of pain, or respiratory abnormality during the training [22].

### 2.3. Sample Size Calculation

This study used the G*power 3.1.9.7 (Franz Faul, University of Kiel, Kiel, Germany) for calculation of the sample size, which was determined on the basis of ability to detect a clinically significant improvement in the outcome measures from a pilot study (least 17 participants for each group), and set the effect size as 1.01, the alpha error as 0.05 and power as 0.80.

### 2.4. Intervention

The purpose of applying CSMT was to improve sense of sitting condition, trunk stability, and lower extremity movement. CSMT was conducted for 30 min, 5 times a week for a total of 6 weeks in this study. For the cognitive sensory-motor program using visual sense, somatosensory, and spatial task as sensory cognitive tasks, an exercise program was set up to distinguish distance and direction. Based on previous studies of Chanubol et al. [5] and Cappellino et al. [23] contents were corrected, supplemented, and consulted with senior physiotherapist and rehabilitation specialists. The program of CSMT training is shown Table 1.

**Table 1.** Cognitive sensory motor training program.

| Intervention | Training Procedure |
|---|---|
| Proprioception and pressure training | Training was performed to improve the sense of position and sense of movement among proprioception senses in the first area and the last two areas after applying pressure to the ankle joint. |
| Tactile and pressure Stimulation training | CSMT was given using contact tasks that presented cognitive problems to distinguish the difference between the degree of pressure and surface material and the difference between sense of friction and weight. It was made possible to distinguish each sense by using visual sense and somatosensory during the task of distinguishing each sense. |
| Tactile and pressure Stimulation training | The trunk that presented cognitive problems was focused on in the training to distinguish the difference in the degree of sponge pressure in the sitting position. It was made possible to distinguish each sense using visual sense and somatosensory during the task of distinguishing each sense. |
| Proprioception and Spatial task training | It was divided into four sections after connecting the knee line of the affected lower limb and the center line to the line to train where the position of the foot of lower limb was located in space. |
| | Two lines were made according to the angle of the knee while the subject's heels were attached. The training was then conducted on the position of the leg in space. If this was undertaken completely, it was performed by adding a line. |
| | To recover spatial cognition, a movement was made to retrace an ellipse in front of the body. From the small position of a drawing board, the movement was induced along an increasingly large circle. |

In order to enhance the effect of exercise training, the training was conducted in an independent place without noise so that an unobstructed environment was provided while maintaining a proper temperature for conducting the training. From the beginning to the end of the study, the training was carried out with the same contents by the same investigator.

For the control group, Bobath's neurodevelopmental therapy and proprioceptive neuromuscular facilitation [24], range of motion (ROM), stretching, upper and lower extremity muscle strength, walking training, and bicycle exercise [25] were given. Subjects underwent general physical therapy for 30 min at a time, twice a day, 5 days a week for 6 weeks.

*2.5. Measurement*

2.5.1. Muscle Strength

Muscle strength of the lower extremity was evaluated using a digital muscular dynamometer (Model 01163, Lafayette Inc., Lafayette, IN, USA, 2003) for the dorsiflexion. The accuracy of the digital muscular dynamometer was 99%. Its intra-rater reliability and inter-rater reliabilities were $r = 0.90\sim0.96$ and $r = 0.76\sim0.97$, respectively [26]. The digital muscular dynamometer can measure the muscle strength of each body part up to 125 lb (56.7 kg). The maximum muscle strength threshold can be determined. For each muscle, the pressure at the time of maximal isometric contraction was measured. The average of three measurements was calculated. A break of 5 s between measurements was given to rule out muscle fatigue. All muscle strength measurements using the dynamometer were recorded using kg. LEMTA was measured by placing a pressure platform on the distal side of the instep [27]. In addition, the lower extremity muscle strength was recorded with the MRC scale using the sum of each score by conducting evaluation of the hip abductor of the affected side, flexor and extensor of the knee joint, and dorsiflexor and plantarflexor of the ankle joint [28]. The evaluation re-evaluation reliability of the MRC scale and the reliability between assessors were $r = 0.84\sim0.96$ and $r = 0.70\sim0.96$, respectively [29].

2.5.2. Balance Test

To measure the balance ability of participants, we used a balance ability measurement and training system (Analysis systems by biofeedback, AP1153 BioRescue, Rodez, France). In the balance assessment, the following four parameters of eye open and close were evaluated: Romberg eye open surface area (REOSA), Romberg eye open average speed (REOAS), Romberg eye close surface area (RECSA), and Romberg eye close average speed (RECAS). They were measured with open eyes in a standing position. Both feet were supported by a force platform. COG movement distance was measured for 60 s. The unit of COG movement distance for 60 s with eyes closed was cm and the unit of surface area was $mm^2$. The smaller the figure from the evaluation, the better was balance ability with little shaking. The limit of stability (LOS) test was performed to measure the COG in eight directions [forward (FW), backward (BW), right (RT), left (LT), forward-right (FW-RT), forward-left (FW-LT), backward-right (BW-RT), and backward-left (BW-LT)] with a unit of $cm^2$. When the subject was standing on the force plate, the direction of the arrow appeared randomly on the computer screen. The subject then moved the COG in the direction of the arrow that appeared randomly on the computer screen. The larger the measured value, the better the dynamic balance ability. All study subjects repeated this three times. The mean value of the three measurements was then calculated.

2.5.3. Statistical Methods

The SPSS (Version 20.0, IBM, Chicago, IL, USA) was used for all statistical analyses of this study. Among general characteristics of the two groups, gender, diagnosis, paralyzed side, spasticity, and encephalopathy area were analyzed by Chi-square test. Homogeneity was analyzed by Independent *t*-test for dependent variables such as age, height, body weight, MMSE-K, and year of onset before training. The normality test was performed with Shapiro–Wilk test. The effects of intervention LEMTA, MRC, and the balance ability test were examined using the two-way repeated measures ANOVA analysis. The pre- and post-test (time) were the within-participants factors. The group-by-time (between-factors) were the CSMT and control group results. When significant differences were observed in group-by-time (main effects or interactions) analyses, the independent-test and paired *t*-test were used post hoc analysis. All statistical significance levels of data were set at $\alpha = 0.05$.

## 3. Results

### 3.1. General Characteristics of Subjects

Table 2 shows general characteristics, medical characteristics, and homogeneity test results of dependent variables.

**Table 2.** General and clinical Characteristics of Subjects ($N$ = 35).

| Variable | CSMT Group ($n$ = 17) | Control Group ($n$ = 18) | $X^2$/t | $p$ |
|---|---|---|---|---|
| Height (cm) | 165.58 (7.62) [a] | 166.54 (5.82) [a] | −0.421 [c] | 0.676 |
| Weight (kg) | 64.42 (8.53) [a] | 65.13 (10.81) [a] | −0.215 [c] | 0.831 |
| Age (year) | 51.75 (14.41) [a] | 58.22 (16.53) [a] | −1.340 [c] | 0.189 |
| Gender (Male/Female) | 9/8 | 10/8 | 0.024 [b] | 0.877 |
| Diagnosis (Infarction/Hemorrhage) | 8/9 | 11/7 | 0.696 [b] | 0.404 |
| Affected side (Left/Right) | 9/8 | 9/9 | 0.030 [b] | 0.862 |
| MMSE-K (score) | 26.65 (1.27) [a] | 26.89 (1.37) [a] | −0.541 [c] | 0.592 |
| Brunnstrom stage | 2.64 (0.49) [a] | 2.72 (0.46) [a] | −0.466 [c] | 0.644 [c] |

[a] Mean(SD). [b] Chi-square test; [c] Independent $t$-test. MMSE = mini-mental state examination. CSMT group = cognitive sensory-motor training group.

### 3.1.1. Comparison of LEMTA and MRC

A Table 3 shows comparison results of LEMTA and MRC between the two groups. Significant between-participant changes were LEMTA ($F$ = 6.760, $p$ = 0.014) and MRC score ($F$ = 96.243, $p$ = 0.000). LEMTA and MRC showed significantly more in the CSMT group than in the control group. Significant within-participant changes were LEMTA ($F$ = 94.053, $p$ = 0.000) and MRC score ($F$ = 380.193, $p$ = 0.000) (Table 3).

**Table 3.** Comparison of LEMTA and MRC between CSMT and Control groups ($N$ = 35).

| Variable | CSMT Group ($n$ = 17) | | Control Group ($n$ = 18) | | $F$ | $p$-Value |
|---|---|---|---|---|---|---|
| | **Pre-Test** | **Post-Test** | **Pre-Test** | **Post-Test** | | |
| LEMTA (kg) | 3.70 (0.54) [1] | 4.07 (0.52) [a,b] | 3.88 (0.68) | 4.10 (0.61) [a] | 6.760 | 0.014 [*,2] |
| MRC (score) | 13.35 (1.58) | 16.88 (1.83) [a,b] | 13.61 (1.72) | 14.78 (1.66) [a] | 96.243 | 0.000 [*,2] |

[1] Mean ± SD. [*] $p$ <0.05, [a] There was a significant difference between pre- and post-test ($p$ < 0.05). [b] The CSMT goup improved more than the control group. [2] Analyed by two-way repeated measures ANOVA, LEMTA, lower extremity muscle strength of tibialis anterior. MRC, medical research council. CSMT group, cognitive sensory-motor training group.

### 3.1.2. Comparison of REOSA and REOAS

Table 4 shows comparison results of REOSA and REOAS between the two groups. Significant between-participant changes were REOSA ($F$ = 10.066, $p$ = 0.003) and REOAS ($F$ = 7.797, $p$ = 0.009). REOSA and REOAS showed significantly more differences in the CSMT group than in the control group. Significant within-participant changes were REOSA ($F$ = 81.201, $p$ = 0.000) and REOAS ($F$ = 236.028, $p$ = 0.000) (Table 4).

### 3.1.3. Comparison of RECSA and RECAS

Table 4 shows comparison results of RECSA and RECAS between the two groups. A significant between-participant changes were RECSA ($F$ = 6.501, $p$ = 0.016) and RECAS ($F$ = 19.372, $p$ = 0.000). RECSA and RECAS showed significantly more differences in the CSMT group than in the control group. Significant within-participant changes were RECSA ($F$ = 74.393, $p$ = 0.000) and RECAS ($F$ = 162.299, $p$ = 0.000) (Table 4).

### 3.1.4. Comparison of LOS

Table 3 shows comparison results of LOS between the two groups. A significant between-participant changes was LOS ($F$ = 5.514, $p$ = 0.025). After training, LOS showed significantly more changes in the CSMT group than in the control group. Significant within-participant changes were LOS ($F$ = 31.544, $p$ = 0.000) (Table 3).

**Table 4.** Comparison of balance ability between CSMT and Control group ($N$ = 35).

| Variable | CSMT Group ($n$ = 17) | | Control Group ($n$ = 18) | | F | $p$-Value |
|---|---|---|---|---|---|---|
| | Pre-Test | Post-Test | Pre-Test | Post-Test | | |
| REOSA (mm$^2$) | 134.12 (39.00) [1] | 122.18 (39.72) [a,b] | 132.00 (36.22) | 126.28 (37.46) [a] | 10.066 | 0.003 *,[2] |
| REOAS (cm/s) | 1.09 (0.28) | 0.89 (0.24) [a,b] | 1.07 (0.28) | 0.93 (0.27) [a] | 7.797 | 0.009 *,[2] |
| RECSA (mm$^2$) | 175.00 (24.99) | 161.24 (26.38) [a,b] | 173.87 (26.74) | 164.72 (28.03) [a] | 6.501 | 0.016 *,[2] |
| RECAS (cm/s) | 1.42 (0.29) | 1.21 (0.28) [a,b] | 1.39 (0.36) | 1.23 (0.37) [a] | 19.372 | 0.000 *,[2] |
| LOS (cm$^2$) | 3086.95 (567.40) | 3507.00 (404.19) [a,b] | 3162.14 (396.33) | 3334.50 (375.42) [a] | 5.514 | 0.025 *,[2] |

[1] Mean ± SD. * $p$ < 0.05, [a] There was a significant difference between pre- and post-test ($p$ < 0.05). [b] The CSMT group improved more than the control group. [2] Analyzed by two-way repeated measures ANOVA, REOSA: Romberg eye open surface area. REOAS: Romberg eye open average speed. RECSA: Romberg eye close surface area. RECAS: Romberg eye close average speed. LOS: limits of stability. CSMT group, cognitive sensory-motor training group.

## 4. Discussion

This study was conducted to investigate effects of CSMT on muscle strength of the lower extremity and balance ability of stroke patients. CSMT was performed for 6 weeks to analyze its effects on lower extremity muscle strength and balance ability of stroke patients.

To our knowledge, this is the first time that CSMT has been applied to the lower extremities in stroke patients. The CSMT method requires a lot of time for the therapist to perform a treatment directly 1: 1 in an independent space. This study was designed to make the patient feel a sensation and to function well cognitively and make exercise and movement better than in an optimal independent space. The therapeutic goal for patients with upper motor-neuron lesion was to make movement for sensory recovery, brain activation, and correct movement input sequence while interacting with each other. According to results of a previous study, the CSMT method can be used in a randomized controlled manner focusing on the recovery of the upper limb function of stroke patients [5]. However, studies about its effects on muscle strength of lower extremity and balance ability have not yet been reported.

The advantage of this study was that the method of applying CSMT to stroke patients with hypoesthesia and dysfunction in lower extremity was a very interesting treatment method. Another important feature was that the cognitive sensory motor method provides various information input to the subjects in three ways: proprioception, somatic sense, and spatial sense. Results confirmed that it was a good method for increasing learning effects as well as motivation by applying various types of senses. The cognitive sensory exercise in this study synchronized muscle contraction with the exercise intention, thus enabling participants to induce the formation of an augmented sensory feedback circuit for performing exercise tasks [30].

The most common exercise impairment after the onset of stroke is weakness of muscle strength. Muscular weakness is a factor that limits functional rehabilitation of stroke patients. It is the target of treatment for performing functional movement. It is an essential factor for improving balance [31]. Saeys et al. [32] reported that trunk exercise was additionally performed in a common general physical therapy, it is more effective for balance and mobility in standing posture because of smooth anticipatory postural performance and harmonization with the extremities by increasing in trunk strength. Derakhshanfar et al. [12]

reported that sensory intervention of exteroceptive and proprioceptive stimulation was effective in improving on motor function and activities of daily living.

The CSMT group showed statistically significant differences up to 6 weeks compared to the control group in the muscle strength of anterior tibialis muscle and MRC scale. This means that the application of CSMT can change the sensory input to the muscle and joint receptors as well as skin receptors on the patient's soles. Such change in sensory input from these receptors can induce the patient's motor system due to motor function rehabilitation [33]. In addition, the improvement of muscle strength can support the body weight effectively because the sensory input of the lower extremity and muscles of the affected lower extremity are activated. Muscle strength is also improved because muscle re-education is achieved by inducing the correct body alignment on the paralysis side.

Balance is the ability to maintain body weight within the base of support with minimal postural sway [34]. The control of balance in stroke patients is a comprehensive process that integrates vestibular, visual, and somatosensory inputs into the central nervous system. Stroke patients have difficulty controlling postural control. Thus, they show an asymmetric posture, balance impairment, and weight shifting [35].

In this study, the CSMT group showed statistically significant differences up to 6 weeks compared to the control group in balance ability. Lim [36] has reported that a multi-sensorimotor training program was effective in improving the proprioception and balance ability for 8 weeks. Liu et al. [37] of 88 chronic stroke patients, it was argued that subjects with cognitive behavioral therapy with task oriented showed increased balance ability and reduced fall risk. Kannan et al. [38] have reported a significant increase in balance control of stroke patients after cognitive-motor exergame training with Wii-fit games. Hong et al. [39] have reported that cognitive task training can significantly improve dynamic balance and gait of stroke patients. Krukowsk et al. [40] have also indicated that the NDT-Bobath and PNF methods show statistically significant difference based on COP movement distance measurements. The balance ability of the chronic stroke patients might improve; this is because the tactile, proprioceptive sense, spatial tasks, and pressure in the CSMT intensively delivered the afferent information in an independent space. Additionally, balance ability may improve by CSMT inducing movements through the sensory and motor systems. Thus, it is demonstrated that the CSMT showed a favorable effect on the improvement of balance ability through this effect.

The intervention of this study strengthened joint mobility, normalization of neuromuscular control, and unconscious motor response to afferent stimulation. Thus, the cognitive motor rehabilitation exercise program was an appropriate and efficient feedforward control so that sensory input information matched current characteristics of the body. It is thought that the balance ability is improved by constantly updating the body's cognitive ability to the most recent state while being adjusted. This study has some limitations. It is difficult to generalize results because the number of participants who participated in this study is too small. Cognitive function, sensory, muscle fatigue, range of motion and spasticity were not tested. Various types of scientific studies focusing on rehabilitation programs for improving lower extremity muscular strength and balance ability in stroke patients need to be carried out in the future.

## 5. Conclusions

CSMT could significant improve muscle strength of lower extremity and balance ability for stroke patients. Thus, it can be applied as a program to increase muscle strength of lower extremity and balance ability of stroke patients who exhibit difficulty in function recovery when planning a training program for stroke patients. Future research may investigate the effects of various interventions combined with CSMT and contribute to integrating sensory, motor, and cognitive effects in stroke patients.

**Author Contributions:** K.-H.K.; conceptualization, S.-H.J.; methodology, K.-H.K., S.-H.J.; investigation, measurement, and writing-review and editing. All authors have read and agreed to the published version of the manuscript.

**Funding:** This work was supported by the National Research Foundation Korea (NRF) grant funded by the Korean government (MSIT) (No. 2019R1G1A1011657).

**Institutional Review Board Statement:** The study was conducted according to the guidelines of the Declaration of Helsinki, and approved by the Institutional Review Board of Sahmyook University (2-1040781-AD-N-01-2016009HR).

**Informed Consent Statement:** Informed consent was obtained from all subjects involved in the study.

**Data Availability Statement:** Not applicable.

**Conflicts of Interest:** The authors declare no conflict of interest.

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
