# Peer review of "Effects of Cognitive Sensory Motor Training on Lower Extremity Muscle Strength and Balance in Post Stroke Patients: A Randomized Controlled Study"

_clinpract, doi:10.3390/clinpract11030079_

Round 1

Reviewer 1 Report

Title: Effects of Cognitive Sensory Motor Training on lower extremity muscle strength and balance in Post Stroke Patients: A Randomized Controlled Study

In this study, the authors investigated the effects of cognitive sensory-motor training (CSMT) on muscle strength and balance ability in post-stroke patients. For stroke patients, the most common neuromuscular condition is sensory-motor impairment, it will be often contributing to muscle weakness and imbalance. This study has chosen a good topic of research in people with Stroke.

But there are the main shortcomings of the study that need minor revision.

  • Abstract
  • Line 11: …. often contributes to muscle weakness and inbalance
  • imbalance? pls, check the spell.
  • Line 19-20: Results: LEMTA (p = 0.014), MRC scale (p = 0.000), balance ability (p < 0.05) were significantly more improved in the CSMT group than in the control group.
  • Match the p-value. ‘p =’ or ‘p <’
  • Introduction
  • Line 65-69: Although there are many papers applying CSMT to the upper extremity for cognitive sensory rehabilitation, papers applying it to the lower extremity have not been published yet.

To improve muscle strength of the lower extremity and balance ability of stroke patients, CSMT might be useful.

  • Two sentences contradict each other. The researcher said there was no article that applied CSMT to lower extremity, but Researcher said ‘CSMT might be useful’.
  • It is necessary to revise the necessary part of the study.
  • Materials and Methods
  • It would be good to present the intervention in a table.
  • Results
  • Add to the value of the Independent t-test or Chi-square test in Table 1.
  • Change ‘MMSE, mini-mental state examination’ to ‘MMSE= mini-mental state examination’ in note of tables.
  • Discussion
  • This section must be improved. A more new and extensive bibliographic review is suggested.
  • Conclusions
  • In addition to what is considered, the strengths and limitations of the study should be reflected. Future prospects must be incorporated.
  • Bibliographic references
  • It must be expanded and updated. I have only found 4 quotes from the last 5 years.

Reviewer 2 Report

This paper is very interesting and important from practical point of view. Perfetti method is very useful in neurorehabilitation. 

Remarks: 

In Discussion/Introduction you should include some information about other similar methods used in neurorehabilitation. Discussion should be more focused on  comparison your results with other articles , and highlighting your best result from this clinical study with practical comments.

You should also mention a few words more about limitation to this study it is not only small group of participants.

Round 2

Reviewer 2 Report

Article is much improved and from my point of view is ready to be published.

This manuscript is a resubmission of an earlier submission. The following is a list of the peer review reports and author responses from that submission.